# Magnetic monopole density and antiferromagnetic domain control in spin-ice iridates

M. J. Pearce[1,2], K. Götze [1,3], A. Szabó[2,4,5], T. S. Sikkenk[4,6], M. R. Lees [1], A. T. Boothroyd [2], D. Prabhakaran[2], C. Castelnovo [4✉] & P. A. Goddard[1✉]

Magnetically frustrated systems provide fertile ground for complex behaviour, including unconventional ground states with emergent symmetries, topological properties, and exotic excitations. A canonical example is the emergence of magnetic-charge-carrying quasi-particles in spin-ice compounds. Despite extensive work, a reliable experimental indicator of the density of these magnetic monopoles is yet to be found. Using measurements on single crystals of $Ho_2Ir_2O_7$ combined with dipolar Monte Carlo simulations, we show that the isothermal magnetoresistance is highly sensitive to the monopole density. Moreover, we uncover an unexpected and strong coupling between the monopoles on the holmium sub-lattice and the antiferromagnetically ordered iridium ions. These results pave the way towards a quantitative experimental measure of monopole density and demonstrate the ability to control antiferromagnetic domain walls using a uniform external magnetic field, a key goal in the design of next-generation spintronic devices.

[1] Department of Physics, University of Warwick, Coventry, UK. [2] Department of Physics, University of Oxford, Clarendon Laboratory, Oxford, UK. [3] Deutsches Elektronen-Synchrotron (DESY), Hamburg, Germany. [4] T.C.M. Group, Cavendish Laboratory, J. J. Thomson Avenue, University of Cambridge, Cambridge, UK. [5] ISIS Facility, Rutherford Appleton Laboratory, Didcot, UK. [6] Institute for Theoretical Physics and Center for Extreme Matter and Emergent Phenomena, Utrecht University, Utrecht, The Netherlands. ✉email: cc726@cam.ac.uk; p.goddard@warwick.ac.uk

Quintessential examples of spin-ice compounds include $Ho_2Ti_2O_7$ and $Dy_2Ti_2O_7$[1–3], where magnetic $Ho^{3+}/Dy^{3+}$ ions sit at the vertices of corner-sharing tetrahedra, which connect to form the pyrochlore structure. The rare-earth moments are constrained by easy-axis anisotropy to point either into or out of each tetrahedron. The resultant ground state is one where two spins point into each tetrahedron and two point out (2I2O), referred to as the 'ice rule' by analogy with proton disorder in water ice[4]. The natural excitations of spin-ice arise from flipping a moment, resulting in one tetrahedron exhibiting a one-in-three-out magnetic moment configuration (1I3O), and the neighbouring tetrahedron arranged three-in-one-out (3I1O)[5–7]. These ice-rule violating defects are understood as deconfined positive and negative magnetic monopoles, respectively[8–11].

In contrast with the non-magnetic Ti atoms of $RE_2Ti_2O_7$ (RE = rare earth), pyrochlore iridates ($RE_2Ir_2O_7$) contain the additional magnetism of the $Ir^{4+}$ ions, which sit on a pyrochlore structure interpenetrating that of the rare-earth moments. The Ir moments are coupled antiferromagnetically and, with the exception of $Pr_2Ir_2O_7$, spontaneously order at low temperatures such that they point alternately all into/all out of adjacent tetrahedra[12]. The effect of this transition on the Ho moments is observed as a slight bifurcation of the field-cooled and zero-field-cooled magnetic susceptibilities[13], which is most clear on subtraction of the two datasets (Fig. 1a). A metal-insulator transition (MIT) occurs concomitantly with the Ir ordering[13]. Measurements of the magnetic susceptibility and resistance show that for single crystals of $Ho_2Ir_2O_7$ the Ir ordering (Fig. 1a) and MIT (Fig. 1b) occur at approximately 80 K, which is significantly higher than any energy scale associated with the magnetism of the

Ho sublattice. The ordered Ir moments produce a local effective magnetic field ($h_{loc}$) at the Ho sites aligned either parallel or antiparallel to the local $\langle 111 \rangle$ directions (see Fig. 1c). These fields, when combined with the spin-ice physics, lead to a fragmented Ho 3I1O/1I3O monopole crystal ground state at sufficiently low temperatures[14].

Spin ices exhibit an anisotropic response to applied magnetic fields. A [100] field stabilises a specific monopole-free 2I2O ground state, whereas a sufficiently large [111] field promotes the formation of magnetic monopoles, arranging the rare-earth moments into a 3I1O/1I3O monopole crystal. The experimental study of this anisotropic behaviour requires measurements on single crystals, something which has only become possible very recently for the pyrochlore iridates[15].

On cooling $Ho_2Ir_2O_7$ through the Ir ordering temperature, two possible Ir antiferromagnetic-domain types form, each of which energetically favours one of the two possible types of Ho 3I1O/1I3O monopole crystal over the other. In this paper we demonstrate that a uniform external field acting on the frustrated Ho moments leads to an energetic pressure that preferentially grows one Ir-domain type over the other. Indeed, a field applied in the [111] direction is known to promote a specific monopole crystal of Ho moments. As this happens uniformly across the sample, it follows that a [111] field favours the Ir-domain type that hosts the compatible monopole crystal order and disfavours the other. This is illustrated in the cartoon in Fig. 1d, with yellow and blue regions representing the two Ir-domain types. Here, circles of the same (different) colour as the background represent monopoles compatible (incompatible) with the underlying Ir order. A [111] field favours monopoles that are compatible with one Ir domain,

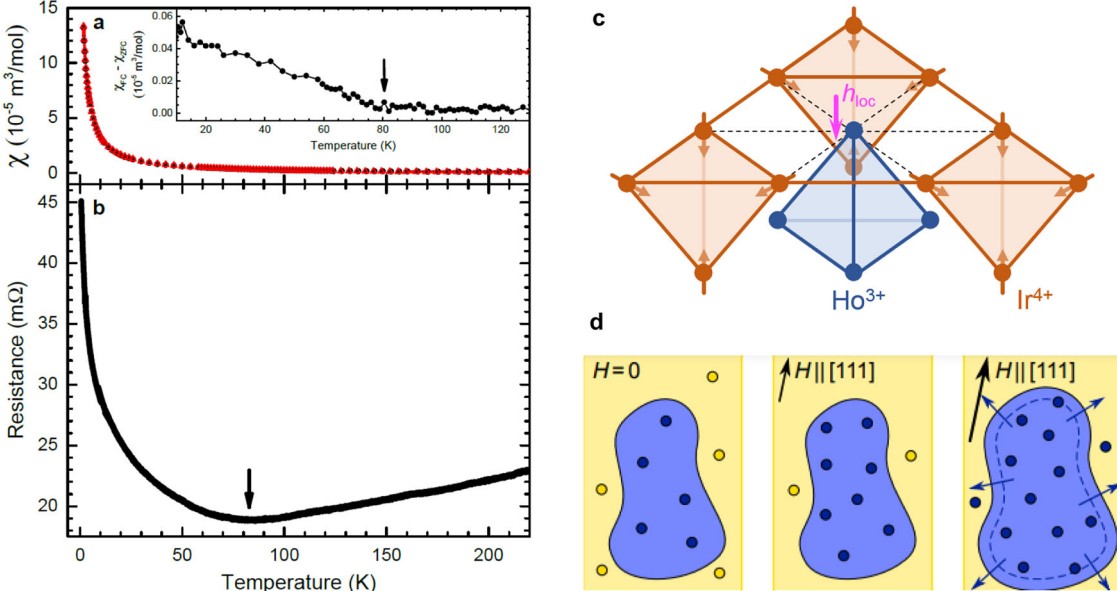

**Fig. 1 Iridium ordering and metal-insulator transition in single-crystalline $Ho_2Ir_2O_7$. a** Field-cooled (red open triangles) and zero-field-cooled (black filled circles) magnetic susceptibility measured in a 0.01 T [100] magnetic field. Inset: Field-cooled magnetic susceptibility with the zero-field-cooled measurement subtracted. The ordering of the Ir moments at approximately 80 K is indicated by the arrow. **b** Resistance measured under zero applied field, exhibiting a MIT at approximately 80 K, indicated by the arrow. **c** Schematic of the pyrochlore structure of $Ho_2Ir_2O_7$, in which each $Ho^{3+}$ ion (blue) has six $Ir^{4+}$ (orange) nearest neighbours. The long-range magnetic order of the Ir moments (orange arrows) results in a net local effective field at the Ho sites ($h_{loc}$) aligned either parallel or antiparallel to the local $\langle 111 \rangle$ directions (indicated here by the magenta arrow for the uppermost Ho site)[14]. **d** Cartoon showing the two possible Ir antiferromagnetic domains (blue/yellow regions) and the different types of Ho monopole crystal order (blue/yellow circles). At zero field, Ho—Ir interactions introduce a finite density of monopoles, consistent with a 3I1O/1I3O monopole crystal, which lowers the local interaction energy (left). Here, the colour of the monopole crystal matches the domain in which it is energetically favoured by the Ho—Ir interaction. An external magnetic field along the [111] direction favours one Ho monopole crystal over the other — say the one in the blue domain, causing the density to increase; in the yellow domain, the density is conversely depleted (centre) but still keeps the domain metastable. A sufficiently strong [111] field, however, repolarises the monopoles in the yellow domain, making it favourable for the blue domain to grow into the yellow one (right).

thus increasing their density, and disfavours monopoles compatible with the other Ir domain, thus decreasing their density and eventually promoting (at sufficiently large fields) a surplus of incompatible monopoles. This then leads to the growth of one Ir antiferromagnetic domain over the other.

This argument is supported by our measurements of the magnetic-field dependence of the magnetisation and resistance of $Ho_2Ir_2O_7$ alongside dipolar Monte Carlo simulations of magnetisation and monopole density. To the best of our knowledge these are the first such experimental measurements on single crystals of this material, which is the reason that this mechanism has not been identified previously. In addition, we also find that the form of the magnetoresistance is strongly linked to the field-induced changes in the monopole density, and we discuss the nature of the coupling that links these two quantities. These observations provide a route to a quantitative experimental measure of the monopole density in spin ices.

## Results

**Applied magnetic field parallel to [100].** First we report the results of magnetisation and resistance measurements on single crystals of $Ho_2Ir_2O_7$ under an applied magnetic field oriented along the [100] direction. The phenomenology for this orientation allows a better understanding and appreciation of the complex physics at play when a field is applied along the [111] direction, presented in the following section.

Figure 2a shows the magnetisation rising rapidly under the [100] field to a saturation magnetisation of 5.8(4) $\mu_B$/Ho, in excellent agreement with the value of 5.77 $\mu_B$/Ho expected from the ice rules for this orientation[16,17]. We note that on sweeping

the field continuously between the positive and negative field limits no hysteresis between the upsweeps and downsweeps is observed (see Fig. 2a inset). Monte Carlo simulations of the magnetisation were performed as described in the methods section. Figure 2b shows simulations for $H\|[100]$ which quantitatively reproduce the behaviour of the measured magnetisation over the studied temperature range.

Figure 2c presents the evolution of the electrical resistance under an applied [100] magnetic field. From a temperature-dependent starting value, there is an initial large negative magnetoresistance which flattens out at higher fields. As with the magnetisation, a continuous sweep of the field between the positive and negative field limits yields negligible hysteresis (see Fig. 2c inset).

The temperature dependence of the zero-field resistance arises due to the insulating nature of $Ho_2Ir_2O_7$ below the MIT at 80 K and is independent of the Ho magnetism. The negative magnetoresistance is caused by a reduction in scattering as the Ho moments order under an applied field. This originates in part from the paramagnetic response of spin ice at fixed monopole density and in part from the suppression of the density of magnetic monopoles at sufficiently large field values. Indeed, Fig. 2d presents Monte Carlo simulations which show how a [100] field acts to suppress the monopole density (defined as the fraction of 3I1O/1I3O tetrahedra) as the Ho moments order into a 2I2O monopole-free magnetic state. The zero-field monopole density is close to 50% due to the competition between the spin-ice physics of the Ho moments (which acts to promote 2I2O tetrahedra) and the local effective magnetic field due to the Ir order (which acts to promote AIAO tetrahedra). In $Ho_2Ir_2O_7$, this coupling makes the energy cost of 2I2O and 3I1O/1I3O

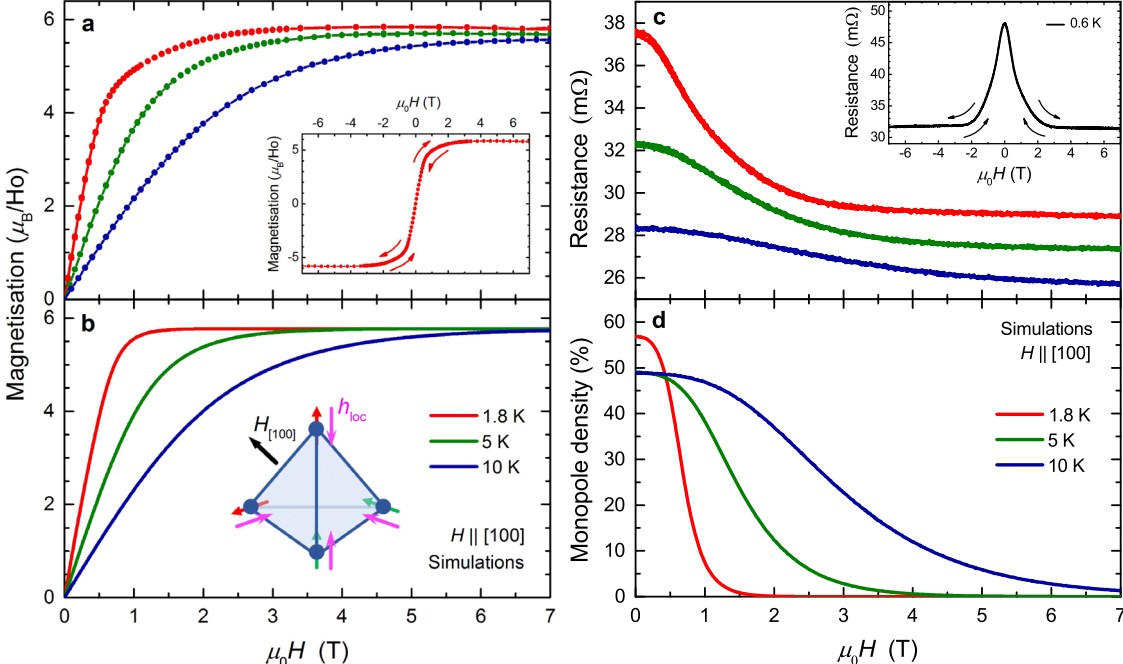

**Fig. 2 $Ho_2Ir_2O_7$ under the application of a [100] magnetic field. a** Measurements and (**b**) Monte Carlo simulations of the magnetisation at various temperatures. The inset to **a** shows a measurement of the magnetisation at 1.8 K upon sweeping the field continuously between the positive and negative field limits. Inset to **b**: a single tetrahedron of the $Ho^{3+}$ sublattice. Magenta arrows indicate the local effective field $\mathbf{h}_{loc}$ due to the ordered Ir moments. Under an externally applied [100] magnetic field (black arrow) the Ho moments order into a 2I2O configuration, oriented either parallel (green arrow) or antiparallel (red arrow) to $\mathbf{h}_{loc}$. **c** Magnetoresistance measurements and (**d**) Monte Carlo simulations of the density of single monopoles (defined as the fraction of 3I1O/1I3O tetrahedra) at various temperatures. Inset to (**c**): a measurement of the resistance at 0.6 K upon sweeping the field continuously between the positive and negative field limits. The temperature dependence of the resistivity is dominated by the insulating behaviour below the MIT, while the negative magnetoresistance at fixed temperature results from a combination of the paramagnetic response of spin ice at fixed monopole density and variations in the monopole density via the mechanisms described in the text.

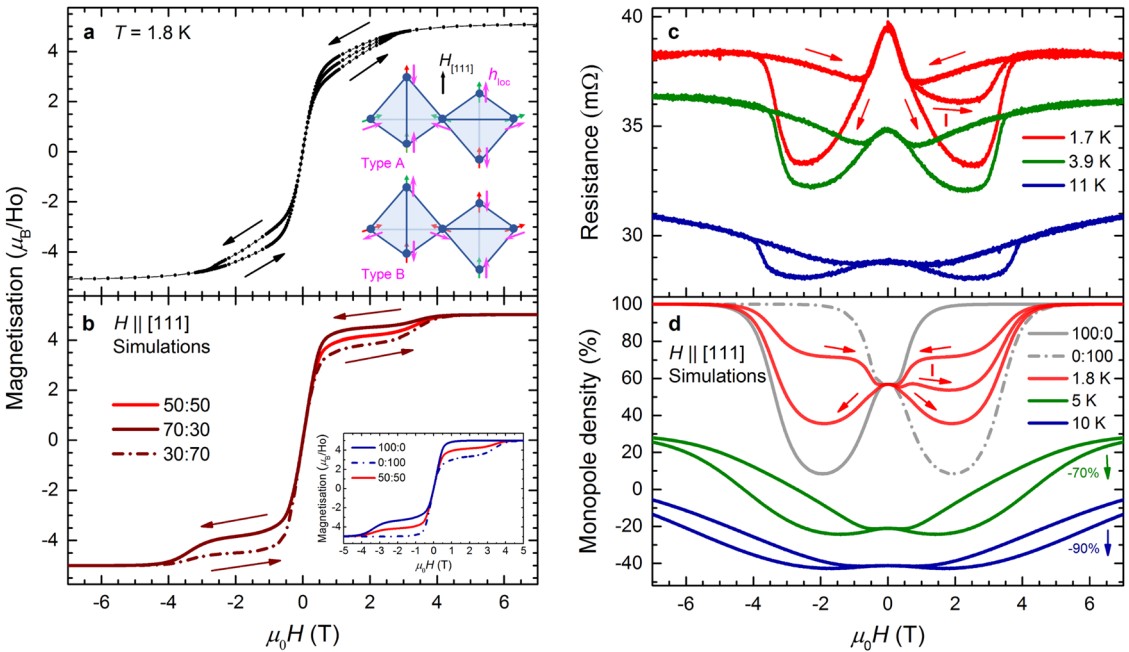

**Fig. 3 Ho₂Ir₂O₇ under the application of a [111] magnetic field. a** Measurements and **b** Monte Carlo simulations of magnetisation at 1.8 K. Arrows in (**a**) indicate the direction of the observed hysteresis loop; the virgin curve lies between the upsweep and downsweep for $H > 0$. Inset to (**a**): tetrahedra of the Ho³⁺ sublattice. Magenta arrows indicate the local effective field $h_{loc}$ for both Ir domain types. An external [111] magnetic field (black arrow) orders the Ho sublattice into a 3I1O/1I3O monopole crystal, with moments either parallel (green arrows) or antiparallel (red arrows) to $h_{loc}$. For this direction of applied field, the alignment favours type-A Ir domains. The 100:0 and 0:100 simulations in the inset to (**b**) represent type-A and type-B single-domain crystals, respectively. Other curves in (**b**) are weighted averages of these. Due to plastic deformation of the Ir domains, the magnetisation of the Ho moments follows the 50:50 curve for the virgin sweep and 70:30/30:70 curves thereafter, indicated by arrows. **c** Magnetoresistance measurements. **d** Monte Carlo simulations of the single monopole density (defined as the fraction of 3I1O/1I3O tetrahedra), simulated using a 50:50 domain ratio for the virgin curve (1.8 K only, labelled 'I') and a 70:30/30:70 average thereafter. Single-domain (100:0/0:100) curves are also shown. The 5 K (10 K) curves in (**d**) are shifted by 70% (90%) to allow for a clearer comparison of the form of the monopole density and experimental magnetoresistance curves (the vertical offset is not relevant to the scattering mechanisms discussed in the text). For higher temperature data see Supplementary Fig. 3.

tetrahedra sufficiently similar to lead to the approximately 50% zero-field value in the temperature range of interest in our work (as noted in Ref. [14], at sufficiently low temperatures Ho₂Ir₂O₇ is expected to undergo a spin-fragmentation transition where the monopoles in fact form a dense crystal). The mechanisms linking resistance and monopole density will be discussed further in the next section.

As the temperature is increased (Fig. 2c), the negative magnetoresistance extends to higher fields. For higher temperatures a larger field is required to order the Ho moments (Fig. 2a) and thus the reduction in spin-dependent scattering occurs over a broader field range. Likewise, higher fields are required to achieve the monopole-free 2I2O state associated with saturation in this orientation, resulting in a slower reduction of the monopole density (Fig. 2d). This behaviour continues for measurements of the magnetisation and resistance above 10 K (see Supplementary Fig. 1).

**Applied magnetic field parallel to [111].** Figure 3a shows that under an applied [111] magnetic field, the magnetisation saturates at 5.1(4) $\mu_B$/Ho, in excellent agreement with the expected value of 5.0 $\mu_B$/Ho for a 3I1O/1I3O spin configuration[16,17].

For this orientation the magnetisation shows a striking hysteresis. At zero applied field the hysteresis loop is closed, indicating that the field sweeps are slow compared to the magnetic response timescale of the Ho moments (see Supplementary Fig. 2). The hysteresis opens at a finite value of applied field: for example, at 1.8 K the hysteresis becomes appreciable around 0.2 T and then closes at 2.9 T. We note that the

temperature range considered in our work is generally well above the end point of the first order transition of spin ice in a [111] magnetic field[18], which therefore cannot be the root of the hysteretic behaviour observed in our measurements. The virgin curve, defined as the first field sweep after cooling from above the MIT in zero magnetic field, sits between the subsequent downsweeps and upsweeps of the hysteresis loop. Other than the virgin sweep, positive and negative quadrants of the magnetisation loop are symmetric within experimental error.

In the Monte Carlo simulations we attempted to induce hysteretic behaviour by increasing the rate at which the magnetic field was ramped, but this necessarily resulted in a hysteresis loop that was open at $H = 0$, in contrast to the experiments. Since single spin-flip Monte Carlo dynamics appears to describe spin-ice materials accurately in the dipolar spin-ice model[19], this finding suggests that the origin of the hysteresis is not intrinsic to the Ho moments and their interactions, a conclusion supported by the fact that the experimental results appear independent of sweep rate (see Supplementary Fig. 2). We are however able to explain the observed hysteresis by proposing that applying a [111] field plastically changes the ratio of Ir-domain types during the course of a field sweep, as we outline below.

For the Ir order that exists below 80 K there are two possible domains, denoted type A and type B (referred to elsewhere as AIAO and AOAI)[15,20,21]. The diagrams in Fig. 3a show the arrangement at the Ho sites of the local effective field produced by the ordered Ir moments for both domain types. The two domains differ by the reversal of the local effective field at each Ho site. An applied [111] magnetic field pushes the initially disordered Ho moments into a 3I1O/1I3O monopole crystal at

saturation[16,17]. This arrangement of Ho moments produces a relative energy saving for a type-A Ir domain due to the favourable alignment of local fields, but leads to a relative energy cost for a type-B Ir domain. By contrast, applying a $[\bar{1}\bar{1}\bar{1}]$ field pushes the Ho moments in the opposite direction and hence favours type-B Ir domains.

If the crystal consisted of a single domain then this effect would lead to an asymmetric magnetisation, as shown in the inset to Fig. 3b, where the 100:0 curve is the calculated magnetisation for a crystal of type-A Ir order only. In this case the external [111] field works in tandem with the local Ir field and the Ho moments saturate rapidly into a 3I1O/1I3O monopole crystal. A $[\bar{1}\bar{1}\bar{1}]$ field ($H < 0$ in the inset) applied to this domain competes with the local Ir field and so must be swept to a higher value to fully align the Ho ions. Also, because of this competition, the Ho moments rearrange via an intermediate regime with vanishing monopole density, resulting in a plateau in magnetisation prior to saturation. The 0:100 curve shows the converse behaviour for a single-domain crystal of type-B Ir order. An average of these two lines is shown in the 50:50 curve, which simulates the response of a crystal containing a fixed and equal ratio of both Ir domains, and is symmetric but not hysteretic. It is clear that none of these situations alone can account for the observed symmetric and hysteretic behaviour; plastic deformation of domain boundaries must be considered.

For the measured magnetisation in Fig. 3a the starting ratio of Ir domain types is expected to be approximately 50:50, because the sample is initially zero-field-cooled through the ordering temperature. It is therefore reasonable that the virgin field sweep closely resembles the 50:50 simulation in Fig. 3b for $H > 0$. As the applied [111] field is swept towards 7 T, an energetic pressure is exerted on the Ir-domain walls. This acts to deform the Ir domains, skewing the ratio in favour of type A, reaching a maximum value once the Ho moments saturate (see Supplementary Note 4). As the applied field is swept from + 7 T to 0 T the energetic pressure is relieved but the A:B ratio does not appear to change, suggesting a plastic deformation has taken place. On sweeping from 0 T to − 7 T the influence on Ir-domain walls is reversed, skewing the ratio now in favour of type-B domains. The ratio again remains fixed as the field is swept from − 7 T to 0 T, before favouring type A once more as the field is increased along [111] and the hysteresis loop is completed.

In support of this picture, we find that subsequent to the virgin curve, the experimental data can be modelled well by a weighted average of the simulated magnetisation for type-A and type-B single-domain crystals in the ratio 70:30 and 30:70 for + 7 T to − 7 T and − 7 T to + 7 T, respectively (see Fig. 3b). This implies that the energetic pressure due to the saturated Ho moments does not force the Ir moments into a single ordered domain at high fields. The most likely explanation is that some form of disorder introduces a distribution of pinning energies for the Ir-domain walls (see Supplementary Notes 4 and 5). This limits the domain imbalance reached, which the comparison between numerical simulations and experiments suggests is approximately 70:30. We note that in our approximate averaging procedure of the Monte Carlo simulations the domain ratio changes abruptly at saturation, whereas experimentally we expect a gradual evolution during the field sweep (see Supplementary Note 4). This approximation likely accounts for the contrast between the smooth experimental curves and step-like behaviour of the simulations. Our proposed mechanism is consistent with the recent experimental results observed for $Dy_2Ir_2O_7$[15], in which the authors comment that the absence of the expected magnetisation plateau could arise from a partially frozen mosaic of iridium domains.

As mentioned earlier, applying a [100] magnetic field orders the Ho spins into a 2I2O spin-ice state. For this configuration, a tetrahedron in either Ir domain contains two Ho moments parallel to the local effective field and two antiparallel (see Fig. 2b inset), giving rise to no energetic pressure between the two Ir domains. Consequently, no hysteresis is expected nor observed. A full study of magnetisation and resistance under an applied [110] field is given in Supplementary Note 6 and provides additional validation for the mechanism proposed here.

Figure 3c shows magnetoresistance in an applied [111] magnetic field which, like magnetisation, is highly hysteretic. The virgin curve (I), shown only for the 1.7 K measurement, lies between subsequent downsweeps and upsweeps. As with the [100] orientation, the form of the magnetoresistance is determined by: (i) the paramagnetic response of spin ice at fixed monopole density, which manifests as a marked drop in resistance at low fields; and (ii) variations of the monopole density as the field is swept.

Figure 3d presents calculations of the density of monopoles for $H \| [111]$. Upon applying a sufficiently large [111] magnetic field the Ho moments order into a 3I1O/1I3O monopole crystal and the monopole density rises to 100%. Supported by the local effective field, this rise is rapid for the favourable Ir domain type (100:0 curve for $H > 0$), while a minimum is present for the less favourable Ir domain (0:100 curve for $H > 0$), as the 3I1O/1I3O state is reached via a regime with vanishing monopole density.

By analogy with the magnetisation analysis, weighted averages of the results for type-A and type-B single-domain crystals are shown, where on moving around the hysteresis loop a 30:70 ratio represents sweeping the field from − 7 T to + 7 T, a 70:30 ratio + 7 T to − 7 T, and a 50:50 ratio the virgin curve (shown for 1.8 K only). Comparing measurements and simulations, it is clear that the resistance and monopole density share a consistent field and temperature dependence (a full discussion of the temperature dependence of both magnetisation and resistance is found in Supplementary Fig. 3). The effect of plastic deformation of the Ir-domain walls is to introduce a notable hysteresis in the monopole density, which reproduces the hysteretic nature of the measured resistance extremely well.

We suggest that the isothermal magnetoresistance and monopole density are linked via two different scattering mechanisms that take place between the conduction electrons and the Ho magnetism at low temperatures. Magnetic scattering occurs between the electronic spin and the magnetic charge associated with a monopole[8]. Furthermore, lattice distortions due to the frustrated magnetic structure generate effective electric dipoles on each Ho tetrahedron hosting a monopole[22,23], resulting in additional scattering of conduction electrons. Both mechanisms are charge-dipole type scattering from emergent monopoles, but via independent magnetic and electric channels. Both effects lead to an electronic scattering rate, and hence a change in resistivity, proportional to the monopole density and sufficiently strong to account for the experimentally observed magnetoresistance (see Supplementary Note 7).

Several mechanisms have previously been proposed to account for various transport properties of spin-ice iridates, especially $Nd_2Ir_2O_7$ and $Pr_2Ir_2O_7$ (see, e.g.,[24–26]). While the mechanisms proposed in these works may be at play in $Ho_2Ir_2O_7$ as well, the much stronger Ho moments mark out dipolar magnetic interactions, and the Coulomb field of spin-ice monopoles in particular, as a preeminent source of scattering in our case. This is also consistent with the close correlation between experimental magnetotransport and calculated monopole densities.

## Discussion

Since the early indirect evidence of magnetic monopoles in spin ice[8], much effort has been devoted to their direct detection and

characterisation, and in particular to measuring their density in experiments. The monopole density relates to thermodynamic and dynamic properties of these materials, and several techniques have been proposed as a measurement proxy, including specific heat[9], neutron scattering[10,11], and magnetic susceptibility and noise measurements[7,19,27–31]. Our results show that the isothermal magnetoresistance of $Ho_2Ir_2O_7$ is strongly linked to the concentration of magnetic monopoles, in a way that holds promise to develop a readily measurable and versatile experimental indicator of their density. Resistance measurements are a fast, straightforward, and widely available experimental technique, which can be performed on very small samples. They also permit time-resolved data collection over a wide temperature range and can be readily combined with high magnetic fields and applied pressure. We note that the scattering mechanisms linking the monopole density and the isothermal magnetoresistance do not depend on the Ir magnetism. Consequently, this technique in principle can be applied to spin-ice systems irrespective of the presence or absence of magnetism at the transition metal site, provided the band gap is sufficiently small to permit resistance measurements. Physical pressure, chemical pressure, and strain in thin films offer routes to alter the band structure, potentially reducing the size of the insulating gap, thus widening the scope of compounds to which this technique may be applied.

Our results demonstrate that the hysteresis in magnetisation and magnetoresistance under an applied [111] magnetic field arises due to a plastic deformation of antiferromagnetic Ir-domain walls. Antiferromagnetic domains are a promising building block for future spintronic devices as they do not produce stray magnetic fields and possess ultrafast spin dynamics[32]. However, manipulating antiferromagnetic domain walls is challenging due to the net-zero magnetisation of the domains and the staggered nature of the field required to interact with the alternating magnetic moments[32]. This is circumvented in $Ho_2Ir_2O_7$ by the interplay between Ir domains and the frustrated ferromagnetism of the large Ho moments, which results in a highly reproducible control over antiferromagnetic domains via a uniform external magnetic field. We also find that the domains are robust to low-level field noise and only fields in excess of approximately 1 T can perturb the magnetic microstructure (see Supplementary Note 4). Whilst the temperature and magnetic field scales associated with domain manipulation in $Ho_2Ir_2O_7$ preclude it from being a suitable candidate spintronic material for everyday devices, our results provide the key ingredients for new materials in which the control of antiferromagnetic domains is possible, namely (i) large frustrated moments; (ii) robust long-range antiferromagnetic order; and (iii) a strong coupling between (i) and (ii). It is intriguing to speculate as to whether such a system operating at higher temperatures may be produced and studied in the realm of artificial spin ice.

Finally we note that recent work on $Bi_2Ir_2O_7$/$Dy_2Ti_2O_7$ heterostructures has addressed the grand challenge of converting spin excitations in a frustrated magnet into an electronic response by observing a connection between spin states in the insulating titanate layer and electronic properties of the non-frustrated iridate layer[33]. Our work brings to light the close links between the magnetic and electric charges in $Ho_2Ir_2O_7$, and hence establishes this interconnectedness in a single material. Similar connections have been predicted to exist theoretically in $Pr_2Ir_2O_7$, but for quite different reasons[34,35]. These links between magnetic and electronic degrees of freedom, together with the inherent interplay we have uncovered between the Ir order and the frustrated ferromagnetism present in the spin-ice iridates, provide new opportunities for the study of complex and out-of-equilibrium behaviour and a framework for possible future functional devices.

## Methods

**Synthesis**. Phase-pure $Ho_2Ir_2O_7$ powder was prepared using high-purity ( > 99.99%) $Ho_2O_3$ and $IrO_2$ with a molar ratio 1:1.05. The 5% excess of $IrO_2$ was added to compensate for evaporation loss. The powder was thoroughly mixed with KF flux in the ratio 200:1 inside an Argon-filled glove box and pressed into 15 mm diameter pellets. The pellets were placed inside a platinum crucible and sintered in a chamber furnace at 1100 °C for 10 h, before cooling to 850 °C at 1 °C/h and finally to room temperature at 60 °C/hour[36]. Octahedral-shaped single crystals were separated after dissolving the flux using hot water. Phase purity of the powder and single crystal samples was characterised using PANalytical and Supernova x-ray diffractometers, respectively. Further details are found in Supplementary Note 8.

**Magnetometry measurements**. The magnetisation ($M$ vs $H$) and susceptibility ($\chi$ vs $T$) of a single crystal of $Ho_2Ir_2O_7$ of mass 0.134(5) mg were measured using a Quantum Design MPMS superconducting quantum interference device (SQUID) magnetometer (with the exception of Supplementary Fig. 2). Magnetisation data at different field sweep rates (Supplementary Fig. 2) were collected using an Oxford Instruments vibrating sample magnetometer (VSM). Demagnetisation effects have been accounted for in the simulations to allow direct comparison with the (uncorrected) experimental data.

**Resistance measurements**. Measurements of the electrical resistance of a single crystal of $Ho_2Ir_2O_7$ of approximate size $0.2 \times 0.2 \times 0.25$ mm$^3$ were made using a four-wire technique with an 855 $\mu$A ac current applied along the [010] direction. We chose to show the experimentally measured resistance of the sample instead of resistivity as the conversion to resistivity is complicated by the small size and the geometry of the samples. An order-of-magnitude estimate of the resistivity for a typical low-temperature, zero-field sample resistance of 35 mΩ is $10^{-5}\Omega$ m (a more detailed discussion can be found in Supplementary Note 7). Magnetic fields were applied using an Oxford Instruments superconducting magnet equipped with a $^3$He insert and were swept at a rate of 1 T/min.

**Simulations**. Monte Carlo (MC) simulations were performed using the full dipolar spin-ice Hamiltonian[37] with an additional local ⟨111⟩ field to represent the coupling to the ordered iridium moments[14]:

$$\mathcal{H} = \frac{J}{3}\sum_{\langle ij\rangle}\sigma_i\sigma_j + D\ell^3\sum_{ij}\sigma_i\sigma_j\left[\frac{\hat{\mathbf{e}}_i \cdot \hat{\mathbf{e}}_j}{r_{ij}^3} - \frac{3(\hat{\mathbf{e}}_i \cdot \mathbf{r}_{ij})(\hat{\mathbf{e}}_j \cdot \mathbf{r}_{ij})}{r_{ij}^5}\right]$$
$$\pm h_{\text{loc}}\sum_i\sigma_i - \mu_{\text{Ho}}\mathbf{B} \cdot \sum_i\sigma_i\hat{\mathbf{e}}_i, \qquad (1)$$

where $J/k_B = -1.56$ K[38] and $D/k_B = 1.34$ K[14] are the strengths of nearest-neighbour exchange and long-range dipolar interactions between holmium spins of magnitude $\mu_{\text{Ho}} = 10\,\mu_B$[14], respectively; $\ell = 3.6$ Å is the distance between nearest-neighbour spins[14]; and $h_{\text{loc}}/k_B = 3.5$ K is the net coupling to the ordered iridium moments, the sign of which depends on the domain type. The value of $h_{\text{loc}}$ was chosen to ensure a good correspondence between the experimental and simulated magnetisation curves in the [100] direction: the difference between our value and the one used in Ref. [14] (6.3 K) likely derives from using the full dipolar Hamiltonian spin-ice model instead of the nearest-neighbour approximation. For the range of fields and temperatures studied, we consider the Ir moments to be fixed in their all-in-all out local directions, similar to[14]. The simulation was performed over $6 \times 6 \times 6$ cubic unit cells of the pyrochlore structure (3 456 spins); periodic boundary conditions for the dipolar interaction were enforced using Ewald summation, including a demagnetising factor for a spherical sample[39] (expected to be a good approximation for our $Ho_2Ir_2O_7$ single crystals, which are truncated octahedra) to allow direct comparison with the (uncorrected) experimental data A sweep rate of 0.2 Oe/MC step was used for all simulations (which we expect to correspond to about 200 Oe/s in real time[19]); we found that all simulations remained in thermodynamic equilibrium.

## Data availability

Data presented in this paper are available at the following archive, http://wrap.warwick.ac.uk/161150/.

## Code availability

The code used to generate the Monte Carlo simulation results shown in the paper is available at https://github.com/attila-i-szabo/Ho2Ir2O7/tree/v1.0 under the GNU General Public License, version 2.

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

## Acknowledgements

We thank T. Orton and P. Ruddy for technical assistance, and E. Lhotel and P. Holdsworth for useful discussions. This project has received funding from the European Research Council (ERC) under the European Union's Horizon 2020 research and innovation program (Grant Agreement No. 681260) (P.A.G.). We acknowledge support from the Engineering and Physical Sciences Research Council (EPSRC) under the following grant numbers: EP/N509796/1 (M.J.P.), EP/P034616/1, EP/M007065/1 and EP/T028580/1 (C.C., T.S.S., and A.S.); and EP/N034872/1 and EP/J017124/1 (D.P. and A.T.B.). A.S., D.P., and A.T.B. acknowledge support from the Oxford-ShanghaiTech collaboration project.

## Author contributions

D.P. and P.A.G. conceived the experiments. A.S. and C.C. conceived and performed the simulations, with initial involvement of T.S.S. D.P. grew the samples. M.P., K.G., M.R.L., and P.A.G. performed the magnetisation and resistivity measurements and analysed the results. M.P., P.A.G., A.S., and C.C. wrote the manuscript with input from all other co-authors. P.A.G., C.C., D.P., and A.T.B. supervised the project.

## Competing interests

The authors declare no competing interests.
