## [Peer Review File · Nature Communications]

REVIEWER COMMENTS

Reviewer #1 (Remarks to the Author):

This paper studies magnetization and magnetoresistance phenomena in the new spin-ice compound $\text{Ho}_2\text{Ir}_2\text{O}_7$. Compared with most previous spin-ice materials, this new compound offers a nice platform to study the interplay between spin-ice physics (on Ho-sublattice), noncoplanar antiferromagnetism (on Ir-sublattice), and itinerant magnetism.

While this work provides further useful information on this material system, I am not sure about the novelty of the results/interpretation. The control of antiferromagnetic domain by magnetic field is not new. And if the authors claim their findings “potentially offering a new avenue for antiferromagnetic spintronic applications”, I suggest they provide some background information, e.g. some relevant references about AFM spintronics? It seems to me that the spin-ice or monopole physics play no direct or decisive role in the AFM domain control. Instead, it is more related to the all-in-all-out noncoplanar AF-order of Ir-spins.

Perhaps the most intriguing result is the reported relation between monopole density (from Monte Carlo simulations) and the magnetoresistance (from experiments). The excellent agreement of the saturation magnetization of Ho-spins indicates that the classical Monte Carlo simulation of the Ho-sublattice spin ice is a reliable tool to study the magnetization behavior. The correlation between the monopole-density and magnetoresistance is obtained for both [001] and [111] field directions.

The explanation of this correlation, however, is not very convincing. The authors attributed the link between resistance and monopole density to magnetic scattering between conduction electron and magnetic monopoles. This essentially treats monopoles as “impurities” or “scatterers” (although monopoles are moving around themselves, they probably can be treated as static impurities for electrons due to the well separated dynamics timescales). Direct application of scattering theory naturally leads to the conclusion that the scattering rate is proportional to monopole density (as outlined in S7 of the supplementary information).

On the other hand, this seems to contradict the temperature dependence of resistance in this regime: As shown in figure 1b, the resistance increases dramatically with decreasing temperature (below the metal-insulator transition at 80K). Yet, the number of monopoles is exponentially suppressed as temperature is lowered. This observation, in my opinion, seems to indicate a more complicated scattering mechanism between conduction electrons and the magnetic sub-system. Such unusual Kondo-like behavior in metallic spin-ice has been investigated in previous works, for

example, Udagawa et al. Phys. Rev. Lett. 108, 066406 (2012), and Chern et al. Phys. Rev. Lett. 110, 146602 (2013). Although it is unclear whether the mechanisms proposed in these previous works can be applied to Ho₂Ir₂O₇, I think it is important to point out the complexity of the electron transport in such systems.

Finally, the calculation in section S7 basically considers electron scattering off an effective dipole-like potential. On the other hand, microscopically the interaction between conduction electron and local magnetic moments is through the on-site Hund's rule coupling as in double-exchange or s-d models. The authors also considered the scattering due to electric-dipole associated with magnetic monopole. It is helpful to discuss (for the benefit of readers) and clarify how to go from the microscopic Hund's coupling to effective dipole potential.

To summarize, I think Ho₂Ir₂O₇ is a nice new compound to study novel spin-ice physics. One of the most important results of this work is the correlation between magnetoresistance and monopole-density. Yet the discussion and interpretation of this phenomenon seem unsatisfactory, I suggest further clarification is required.

Reviewer #2 (Remarks to the Author):

This is a truly impressive study of the magnetization and magnetoresistance of the spin ice material, Ho₂Ir₂O₇. The authors have taken very complete data sets and conducted simulations with which to interpret the results. The topic of connecting spin ice magnetic excitations with conductivity is very interesting, and this study will be of considerable interest and impact.

This paper could be published in Nature Communications, but I have two broad concerns and several more detailed concerns that I believe must be addressed first.

Broad Concerns:

1. The narrative of the data and their interpretation is rather convoluted. In its most condensed form, I might paraphrase it as: "The antiferromagnetic Ir moment domains affect the Ho moments'

response to a field, the Ho moment response is in the form of changing the monopole density, and the change in monopole density seems to correlate well with the magnetoresistance, for which there is good reason to believe connections with the monopoles. This is evidenced by measurements of the magnetization and magnetoresistance, and comparing them with simulations that allow a detailed measure of the monopole density." This story line is very hard to follow, and it obscures the beauty of their results. I would encourage the authors to try to simplify the story to make it easier to follow (if my summary above is incorrect, then that is more evidence of the need for clarity). One suggestion is that I might try explaining the ideas separately from the data, perhaps with some schematics or cartoons to indicate how the Ir moment AFM domains interact with the Ho moments, and thus with the monopoles, and thus with the carriers. This could be before showing the data, and could also differentiate between the two different field directions.

2. I am a bit troubled by the use of the term "monopole density". Is that simply the number of tetrahedra with net magnetic charge? Does it include both mobile and static monopoles? Does the mobility of the monopoles affect the electron-scattering? I would expect such scattering to have temperature dependence if so. Why is it 50% at zero field (I would have expected the system to be near the perfect 2I2O state)? For aficionados of monopoles and spin ice especially, the authors should include a bit more detail to the definition and how the density was obtained.

More detailed concerns:

1. The connection to spintronics is a bit hard to swallow. The physics of the paper is beautiful, but it is physics of spin ice, which only works at low temperatures and depends on very strong magnetic fields that would be impractical in any device. I would either drop this reference, or weaken it, or justify it much more strongly. While I can't think of one, perhaps there is some connection with artificial spin ice systems that could be elaborated upon?

2. The resistivity measurements should all be given in Ohm-cm, rather than simply in Ohms, since that gives a point of comparison for sample quality. The current direction relative to the crystal axis should also be given.

3. Given that the conductivity of this materials is marginal, it could be important to future studies to know the sample quality as well as possible. What is the stoichiometry (or the limit of the knowledge

of the stoichiometry)? Can anything be said about the crystalline quality or defects? Given that a 2% off-stoichiometry in oxygen could impact the valence state of 7% of the Ho or Ir, and thus the magnetic properties, this is potentially important to know for future studies.

Reviewer #3 (Remarks to the Author):

The results presented in the manuscript are sound and provide new paths for both the spintronics and frustrated magnetism community to pursue. The link between magnetic monopole density and resistance in Ho₂Ir₂O₇ which is shown here is well-established by these measurements, and the authors have very plausible explanations for all the features seen in the magnetization and resistance data. The connection between the Ir sublattice magnetic domains and the monopoles from exciting the Ho sublattice spin ice configurations (controlled by a magnetic field) is interesting in light of fundamental results of frustration, but, as the authors creatively suggest, could potentially be used for control of AFM domains for spintronics. I recommend publication in Nature Communications.

My one request is that the authors take one or two lines to explain what results on Dy₂Ir₂O₇ their proposed mechanism is consistent with. They mentioned that it was consistent, and have cited a paper (reference 15), but did not directly explain what the results were, or how the mechanism fits with them.

We thank the referees for their useful input. We are pleased that the reviewers think the paper is "a truly impressive study" and provides "new paths for both the spintronics and frustrated magnetism community to pursue". We are able to address all of the referees' comments and have made a number of changes to the manuscript as a result. The details of all the changes are outlined in turn below.

Comments of Referee 1

"This paper studies magnetization and magnetoresistance phenomena in the new spin-ice compound Ho₂Ir₂O₇. Compared with most previous spin-ice materials, this new compound offers a nice platform to study the interplay between spin-ice physics (on Ho-sublattice), noncoplanar antiferromagnetism (on Ir-sublattice), and itinerant magnetism.

While this work provides further useful information on this material system, I am not sure about the novelty of the results/interpretation. The control of antiferromagnetic domain by magnetic field is not new. And if the authors claim their findings "potentially offering a new avenue for antiferromagnetic spintronic applications", I suggest they provide some background information, e.g. some relevant references about AFM spintronics? It seems to me that the spin-ice or monopole physics play no direct or decisive role in the AFM domain control. Instead, it is more related to the all-in-all-out noncoplanar AF-order of Ir-spins."

The AFM domain control we report is only possible due to the interplay of the ordered Ir domains and the spin-ice magnetism of the Ho moments. The direct coupling between the antiferromagnetic Ir domains and an external magnetic field is negligible in our system. It is the coupling to the Ir domains mediated via the large Ho moments (and in particular the 3I1O/1I3O monopole-hosting tetrahedra associated with a [111] magnetic field) which allows this manipulation of the domains.

To make this point more clear in the manuscript, we have added a paragraph to the end of the Introduction Section and an extra panel to Figure 1, in which we give an introductory overview of the mechanism prior to explaining the details in the results sections, with reference to the magnetisation and resistivity data.

Building on this, in Supplementary Section S4A we discuss the equilibrium thermodynamics of the Ho spin ice coupled to the Ir domains (through the h_{loc} local-field term). Here we demonstrate in detail how the magnetic-field evolution of the effective energetic pressure applied by the spin ice on the domain walls is crucial to explain both the control of the antiferromagnetic domains and the observed stability of the domains against weak magnetic fields. This additional strong support for the role of spin ice in the domain control is now referred to in Section IIB and III of the main text.

Of course, as the referee points out, the Ir AFM ordered state is indispensable to this phenomenon: beyond giving rise to the domains themselves, its interaction with the Ho moments influences the spin-ice thermodynamics and brings about the observed behaviour.

In the penultimate paragraph of the Conclusions Section, we now point out that, although the low temperatures at which domain manipulation occurs in Ho₂Ir₂O₇ preclude it from being a suitable candidate spintronic material for everyday devices, our results provide a guide to how this control could be achieved in the future. Here we also reference a recent review article titled "Antiferromagnetic Spintronics" that highlights the benefits of controlling AFM domains in the context of spintronics and the main experimental obstacles that remain. We believe that our results add a great deal to this discussion.

"Perhaps the most intriguing result is the reported relation between monopole density (from Monte Carlo simulations) and the magnetoresistance (from experiments). The excellent agreement of the saturation

magnetization of Ho-spins indicates that the classical Monte Carlo simulation of the Ho-sublattice spin ice is a reliable tool to study the magnetization behavior. The correlation between the monopole-density and magnetoresistance is obtained for both [001] and [111] field directions.

The explanation of this correlation, however, is not very convincing. The authors attributed the link between resistance and monopole density to magnetic scattering between conduction electron and magnetic monopoles. This essentially treats monopoles as “impurities” or “scatterers” (although monopoles are moving around themselves, they probably can be treated as static impurities for electrons due to the well separated dynamics timescales). Direct application of scattering theory naturally leads to the conclusion that the scattering rate is proportional to monopole density (as outlined in S7 of the supplementary information).

On the other hand, this seems to contradict the temperature dependence of resistance in this regime: As shown in figure 1b, the resistance increases dramatically with decreasing temperature (below the metal-insulator transition at 80K). Yet, the number of monopoles is exponentially suppressed as temperature is lowered. This observation, in my opinion, seems to indicate a more complicated scattering mechanism between conduction electrons and the magnetic sub-system. Such unusual Kondo-like behavior in metallic spin-ice has been investigated in previous works, for example, Udagawa et al. Phys. Rev. Lett. 108, 066406 (2012), and Chern et al. Phys. Rev. Lett. 110, 146602 (2013). Although it is unclear whether the mechanisms proposed in these previous works can be applied to Ho₂Ir₂O₇, I think it is important to point out the complexity of the electron transport in such systems.”

We thank the referee for pointing out these possible alternative approaches and we have now raised these issues in the manuscript. However, we should make it clear that Ho₂Ir₂O₇ differs from Pr₂Ir₂O₇ and high-pressure Nd₂Ir₂O₇, which are the focus of those works, in three very important respects.

- In the latter materials, there is no metal-to-insulator plus AFM transition (MIT) of the kind seen in Ho₂Ir₂O₇. The concomitance of the two transitions in Ho₂Ir₂O₇, as well as the large separation of holmium and iridium energy scales (the transition temperature is about 50 times larger than the monopole gap) strongly suggests that the MIT and the rise of resistivity below it are due to Ir, rather than Ho, magnetism (see, for example, Ueda et al., Phys. Rev. B, 93, 245120 (2016)).
- The interaction between rare-earth moments and the Ir conduction electrons is treated as a short-range Kondo coupling in those papers. This is perfectly reasonable for Pr and Nd, whose moments are relatively small. The significantly larger Ho moments, on the other hand, give rise to dipolar interactions that dominate nearest-neighbour exchange (indeed, this is crucial for stabilising the spin-ice phase in Dy₂Ti₂O₇ and Ho₂Ti₂O₇ despite *antiferromagnetic* exchange). Likewise, Kondo coupling will be significantly smaller than the Zeeman coupling between conduction-electron spins and the magnetic field of these dipoles. The latter can be treated using the dumbbell model of dipolar spin ice (see Castelnovo et al., Nature 451, 42 (2008)): the monopole-free background leads to relatively weak and short-range quadrupolar fields, while monopoles appear as longer range Coulombic $1/r$ sources, and as such will be the dominant sources of scattering. The qualitative difference between long-range dipolar and short-range Kondo coupling may well account for their apparently contrary effect on resistivity (increasing and decreasing it, respectively).

- Our results (which deal with field-induced changes in monopole density at a fixed temperature) are not necessarily comparable to the temperature-dependence computed in the references.

In future work, we plan to study the magnetic scattering mechanism in more detail than is possible here and will include a detailed comparison of the two scattering routes. Motivated by this comment from the referee, we have added a paragraph to the end of Supplementary Section S7 detailing the discussion above regarding Kondo coupling and addressing the complexity of the electron transport in these systems.

“Finally, the calculation in section S7 basically considers electron scattering off an effective dipole-like potential. On the other hand, microscopically the interaction between conduction electron and local magnetic moments is through the on-site Hund’s rule coupling as in double-exchange or s-d models. The authors also considered the scattering due to electric-dipole associated with magnetic monopole. It is helpful to discuss (for the benefit of readers) and clarify how to go from the microscopic Hund’s coupling to effective dipole potential.”

As discussed above, the dipole potential is *not* due to Hund's rule coupling, neither is it effective; it comes from the dipolar magnetic field of the large Ho moments, which are a crucial part of dipolar spin-ice physics. Furthermore, the Hund's rule coupling picture of the references discussed above is slightly oversimplified, as the Ir conduction electrons and the rare-earth moments live on different sites; however, we do not think this has any bearing on their long-wavelength results.

The electric-dipole scattering discussed in our manuscript is due to an entirely separate mechanism; the origins of the electric dipoles in question are explained in detail in Khomskii, Nat. Comms. 3, 904 (2012).

Comments of Referee 2

“This is a truly impressive study of the magnetization and magnetoresistance of the spin ice material, Ho₂Ir₂O₇. The authors have taken very complete data sets and conducted simulations with which to interpret the results. The topic of connecting spin ice magnetic excitations with conductivity is very interesting, and this study will be of considerable interest and impact.

This paper could be published in Nature Communications, but I have two broad concerns and several more detailed concerns that I believe must be addressed first.

Broad Concerns:

1. The narrative of the data and their interpretation is rather convoluted. In its most condensed form, I might paraphrase it as: "The antiferromagnetic Ir moment domains affect the Ho moments' response to a field, the Ho moment response is in the form of changing the monopole density, and the change in monopole density seems to correlate well with the magnetoresistance, for which there is good reason to believe connections with the monopoles. This is evidenced by measurements of the magnetization and magnetoresistance, and comparing them with simulations that allow a detailed measure of the monopole density." This story line is very hard to follow, and it obscures the beauty of their results. I would encourage the authors to try to simplify the story to make it easier to follow (if my summary above is incorrect, then that is more evidence of the need for clarity). One suggestion is that I might try explaining the ideas separately from the data, perhaps with some schematics or cartoons to indicate how the Ir moment AFM domains interact with the Ho moments, and thus with the monopoles, and thus with the carriers. This could be before showing the data, and could also differentiate between the two different field directions.”

We thank the referee for their helpful comment. We have amended two paragraphs at the end of the introduction which now give an overview of the complete storyline. We have also acted on the referee's suggestion by adding the cartoon below as Figure 1(d), which presents the ideas behind how the Ho-moment monopoles and Ir-moment AFM domains interact with each other under the influence of an external magnetic field, prior to the presentation of the details of the experimental data and simulations. We believe that, together, these changes aid the clarity of the manuscript by presenting the key storyline clearly before the data are presented.

Caption: “Cartoon showing the two possible Ir antiferromagnetic domains (blue/yellow regions) and the different types of monopole crystal order (blue/yellow circles). Ho-Ir interactions introduce a finite density of monopoles, consistent with a 3I1O/1I3O monopole crystal, which lowers the local interaction energy (left). Here, the colour of the monopole crystal matches the domain in which it is energetically favoured by the Ho–

Ir interaction. An external magnetic field along the [111] direction favours one monopole crystal over the other – say the one in the blue domain, causing the density to increase; in the yellow domain, the density is conversely depleted (centre) but still keeps the domain metastable. A sufficiently strong [111] field, however, repolarises the monopoles in the yellow domain, making it favourable for the blue domain to grow into the yellow one (right).”

Amended paragraphs at end of Introduction: “On cooling Ho₂Ir₂O₇ through the Ir ordering temperature, two possible Ir antiferromagnetic-domain types form, each of which energetically favours one of the two possible types of 3I1O/1I3O monopole crystal over the other. In this paper we demonstrate that a uniform external field acting on the frustrated Ho moments leads to an energetic pressure that preferentially grows one Ir-domain type over the other. Indeed, a field applied in the [111] direction is known to promote a specific monopole crystal of Ho moments. As this happens uniformly across the sample, it follows that a [111] field favours the Ir-domain type that hosts the compatible monopole crystal order and disfavours the other. This is illustrated in the cartoon in Figure 1d, with yellow and blue regions representing the two Ir-domain types. Here, circles of the same (different) colour as the background represent monopoles compatible (incompatible) with the underlying Ir order. A [111] field favours monopoles that are compatible with one Ir domain, thus increasing their density, and disfavours monopoles compatible with the other Ir domain, thus decreasing their density and eventually promoting (at sufficiently large fields) a surplus of incompatible monopoles. This then leads to the growth of one Ir antiferromagnetic domain over the other.

This argument is supported by our measurements of the magnetic-field dependence of the magnetisation and resistance of Ho₂Ir₂O₇ alongside dipolar Monte Carlo simulations of magnetisation and monopole density. To the best of our knowledge these are the first such experimental measurements on single crystals of this material, which is the reason that this mechanism has not been identified previously. In addition, we also find that the form of the magnetoresistance is strongly linked to the field-induced changes in the monopole density, and we discuss the nature of the coupling that links these two quantities. These observations provide a route to a quantitative experimental measure of the monopole density in spin ices.”

“2. I am a bit troubled by the use of the term "monopole density". Is that simply the number of tetrahedra with net magnetic charge? Does it include both mobile and static monopoles? Does the mobility of the monopoles affect the electron-scattering? I would expect such scattering to have temperature dependence if so. Why is it 50% at zero field (I would have expected the system to be near the perfect 2I2O state)? For aficionados of monopoles and spin ice especially, the authors should include a bit more detail to the definition and how the density was obtained.”

Yes, the referee is correct: the “monopole density” is the fraction of 3I1O/1I3O tetrahedra, i.e. those hosting single magnetic monopoles. To clarify this explicitly for the reader, we have added the phrase “defined as the fraction of 3I1O/1I3O tetrahedra” to the captions to Figures 2 and 3 as well as in the Results Section where this term is first used.

We do neglect the mobility of monopoles. We believe that this is reasonable given the very different dynamical time scales of Ho and Ir (milliseconds and femtoseconds, respectively). We have added a sentence to this effect to Supplementary Section S7, where we discuss the details of the mechanisms linking the monopole density and resistance.

Regarding the value of the zero-field monopole density, the local effective magnetic field due to the AIAO Ir moments aims to order the Ho moments into an AIAO configuration, thereby introducing monopoles. This interaction is in competition with the spin-ice physics of the Ho moments themselves which prefers a 2I2O state. In Ho₂Ir₂O₇, this coupling makes the energy cost of 2I2O and 3I1O/1I3O tetrahedra sufficiently similar to lead to an approximate 50% density of single monopoles (double monopoles remain negligibly rare) at temperatures above the monopole crystallisation (spin fragmentation) transition (which is expected to occur at low temperature, according to Ref [14]). We have added the following explanation to the [100] results section:

“The zero-field monopole density is close to 50% due to the competition between the spin-ice physics of the Ho moments (which acts to promote 2I2O tetrahedra) and the local effective magnetic field due to the Ir order (which acts to promote AIAO tetrahedra). In Ho₂Ir₂O₇, this coupling makes the energy cost of 2I2O and 3I1O/1I3O tetrahedra sufficiently similar to lead to the approximately 50% zero-field value in the temperature range of interest in our work (as noted in Ref. [14], at sufficiently low temperatures Ho₂Ir₂O₇ is expected to undergo a spin-fragmentation transition where the monopoles in fact form a dense crystal).”

“More detailed concerns:

1. The connection to spintronics is a bit hard to swallow. The physics of the paper is beautiful, but it is physics of spin ice, which only works at low temperatures and depends on very strong magnetic fields that would be impractical in any device. I would either drop this reference, or weaken it, or justify it much more strongly. While I can't think of one, perhaps there is some connection with artificial spin ice systems that could be elaborated upon?”

We certainly agree that the temperature and magnetic field scales associated with our system are incompatible with a practical spintronic device. However, we do believe that our results provide a recipe for materials in which the control of antiferromagnetic domains via a uniform external magnetic field is possible. We also think this is something worth pointing out in the manuscript. We have added the following clause to the Conclusions Section to explicitly stress the practical limitations of Ho₂Ir₂O₇:

“Whilst the temperature and magnetic field scales associated with domain manipulation in Ho₂Ir₂O₇ preclude it from being a suitable candidate spintronic material for everyday devices,...”.

We thank the referee for their interesting suggestion regarding artificial spin-ice systems. Currently we do not feel able to make a firm statement on this topic in the manuscript, however we agree that it is worth mentioning and have added the following sentence to the conclusions:

“It is intriguing to speculate as to whether such a system operating at higher temperatures may be produced and studied in the realm of artificial spin ice.”.

“2. The resistivity measurements should all be given in Ohm-cm, rather than simply in Ohms, since that gives a point of comparison for sample quality. The current direction relative to the crystal axis should also be given.”

This is something we thought about carefully prior to submission. However, we have taken the deliberate choice to display the resistance. This is because the conversion to resistivity is complicated by the small size and the geometry of the single crystals. The samples have the shape of a truncated octahedron with an edge length of ~200 μm , and the exact shape and dimensions of the cross-section between the electrical contacts is difficult to accurately determine. Nevertheless, we certainly agree that an indication of the resistivity allows for a better comparison between different samples. As such we have added the following sentence to the Methods Section to provide an order of magnitude estimate of the resistivity, noting the difficulties in doing so:

“We chose to show the experimentally measured resistance of the sample instead of resistivity as the conversion to resistivity is complicated by the small size and the geometry of the samples. An order-of-magnitude estimate of the resistivity for a typical low-temperature, zero-field sample resistance of 35 $\text{m}\Omega$ is $10^{-5} \Omega\text{m}$ (a more detailed discussion can be found in Supplementary Section S7).”

We have also provided further details in Section S7 of the supplementary information.

The current was applied along the [010] direction. We have added a sentence to the Methods Section to indicate this.

“3. Given that the conductivity of this materials is marginal, it could be important to future studies to know the sample quality as well as possible. What is the stoichiometry (or the limit of the knowledge of the stoichiometry)? Can anything be said about the crystalline quality or defects? Given that a 2% off-stoichiometry in oxygen could impact the valence state of 7% of the Ho or Ir, and thus the magnetic properties, this is potentially important to know for future studies.”

Potential problems with the stoichiometry were mitigated at the synthesis stage as follows: we prepared the phase pure polycrystalline Ho₂Ir₂O₇ powder sample with an excess of 5% IrO₂ powder in order to compensate the evaporation loss due to high temperature synthesis and used the final single-phase powder as a starting material for the crystal growth. The growth was carried out at relatively low temperature and did not observe any IrO₂ evaporation. The cell parameter values of both powder and single crystals are consistent and so we would not expect any variation in the chemical composition. Moreover, both Ho and Ir have stable valency states, 3+ and 4+ respectively, so we believe that any significant oxygen variation in

the crystal is unlikely. To confirm this, we have annealed the as grown crystal in the oxygen atmosphere and found no change in the cell parameter value.

Furthermore, a qualitative estimate of the sample quality can be deduced from various physical properties. The measured lattice parameters of our crystals of Ho₂Ir₂O₇ are in excellent agreement with the structural trend across the RE₂Ir₂O₇ family, see Clancy *et al.*, Phys. Rev. B **94**, 024408 (2016) and Subramanian *et al.*, Prog. Solid St. Chem. **15**, 55 (1983). Moreover, it has been previously argued (Sala *et al.*, Nature Materials **13**, 488 (2014)) that the effect of oxygen vacancies in non-Kramers spin-ice materials is to suppress the moments on the rare-earth sites. If this was happening in our crystals of Ho₂Ir₂O₇, it would be detectable via experimental measurements of the saturation magnetization along different crystallographic directions. The results we obtain are in very good agreement with the values expected from calculations made with the full Ho³⁺ moment. While we are sorry that we are unable to offer the referee a quantitative estimate of the sample stoichiometry, we believe that the care taken at synthesis and these experimental markers of structural and magnetic defects strongly imply that our crystals are indeed of a high quality, and at least comparable to other previously published spin-ice samples, either powder or single crystal.

Comments of Referee 3

“The results presented in the manuscript are sound and provide new paths for both the spintronics and frustrated magnetism community to pursue. The link between magnetic monopole density and resistance in Ho₂Ir₂O₇ which is shown here is well-established by these measurements, and the authors have very plausible explanations for all the features seen in the magnetization and resistance data. The connection between the Ir sublattice magnetic domains and the monopoles from exciting the Ho sublattice spin ice configurations (controlled by a magnetic field) is interesting in light of fundamental results of frustration, but, as the authors creatively suggest, could potentially be used for control of AFM domains for spintronics. I recommend publication in Nature Communications.

My one request is that the authors take one or two lines to explain what results on Dy₂Ir₂O₇ their proposed mechanism is consistent with. They mentioned that it was consistent, and have cited a paper (reference 15), but did not directly explain what the results were, or how the mechanism fits with them.”

In this Reference, the authors note that the absence of expected magnetization plateau in Dy₂Ir₂O₇ could be explained by the “presence of a partially frozen mosaic of “AIAO / AOAI” iridium domains (as observed in Nd₂Ir₂O₇ [47]) which drive domains of monopole crystal order”. Our results revealing the indirect connection between the applied field and the Ir AFM domains, mediated by the Ho spin ice, is consistent with this comment. We have expanded on this in Section IIB with the following sentence:

“Our proposed mechanism is consistent with the recent experimental results observed for Dy₂Ir₂O₇ [15], in which the authors comment that the absence of the expected magnetization plateau could arise from a partially frozen mosaic of iridium domains.”

REVIEWER COMMENTS

Reviewer #1 (Remarks to the Author):

The authors have mostly satisfactorily answered my comments/questions, except one point regarding the interpretation of the conductivity and magnetoresistance. Since this is one of the main conclusions, regardless of other new results or novel conclusions, I think the work is not internally self-consistent. It is my strong recommendation that the authors need to more carefully address this issue to be discussed below.

The main issue here is, again, the theoretical interpretation of the resistivity. As summarized in the last sentence of Section II on page 11:

“Both effects lead to an electronic scattering rate, and hence a change in resistivity, proportional to the monopole density and sufficiently strong to account for the experimentally observed magnetoresistance”.

Coupled with their discussion in Supplemental Section S7, the authors claim that the observed change in resistivity is mainly related to “monopole density” (treating monopoles as effective scatterers). However, as I pointed out in previous report, this picture cannot explain the observation that resistivity also increases when the monopole density is reduced by lowering temperature. And the authors did not directly address this comment in their reply, they said:

“Our results (which deal with field-induced changes in monopole density at a fixed temperature) are not necessarily comparable to the temperature-dependence computed in the references.”

Here the “references” are the two papers I mentioned in previous report. However, the point is that their interpretation of “field-induced changes in monopole density at a fixed temperature” is **not** consistent with their own experimental result shown in Fig. 1b and Fig. 2, which clearly shows the resistance increases at low temperatures ($\sim 50\text{K}$, which is also the regime of their field-dependence study). In particular, in their Fig. 2, for magnetic field larger than $\mu_0 H > \sim 0.5\text{ T}$, the monopole density $n_m(10\text{K}) > n_m(5\text{K}) > n_m(1.8\text{ K})$ (shown in panel d). On the other hand, the resistance $R(10\text{ K}) < R(5\text{K}) < R(1.8\text{ K})$ (shown in panel c), which shows the opposite trend.

(By the way, it is a bit misleading that in their Fig. 3d, the authors shifted the “monopole-density vs H” curves for the T = 5K and 10K data downward by 70% and 90%.)

Since the authors claim the resistivity is mainly affected by the “monopole density” (at least in the regime of their study), it shouldn't matter whether the change of monopole density is due to H-field or temperature. The interpretation should work for both H- and T-effect.

It seems to me that in order for their theory to work (resistance is proportional to monopole density), one needs to introduce a strong temperature-dependent factor in the scattering theory. But no such dependence can be seen in Eq. (S4) or (S5) in the supplemental material. Another possibility is that other important scattering mechanism is not properly considered here. And if this is the case, it is unclear how much of their results can be solely attributed to monopole density.

Reviewer #2 (Remarks to the Author):

I am grateful to the authors for fully addressing the comments in my previous review, and I believe the paper is suitable for publication in Nature Communications after one small change.

My only further suggestion is that the authors should include the details of their sample characterization (regarding stoichiometry) in either the methods or the SI. It will be useful for other workers in the field to understand what has been done in this regard -- the issue should be of broad interest. The description that they included in the referee response would be sufficient and could be simply pasted into the SI or Methods section with minimal rewording.

We thank the reviewers for engaging with our response to their earlier comments and with the resulting changes we made to the manuscript. We are pleased to see that Reviewer 2 recommends publication barring one last small change that we have now readily implemented. Reviewer 1 is satisfied with our responses except for one last important point; we thank the reviewer for their patience in further clarifying the criticism in their new report and allowing us to finally address it fully and thoroughly.

We feel that we have now met all the requirements, and in the process greatly improved the manuscript and the clarity of our results and claims. We believe that our work now meets the criteria to be published in Nature Communications.

In this document we detail our answers to the two remaining queries of the referees in turn. (Reviewer comments are in blue, our response in black and **changes to the manuscript in red**.)

Comments of Reviewer #1

*The authors have mostly satisfactorily answered my comments/questions, except **one point regarding the interpretation of the conductivity and magnetoresistance**. Since this is one of the main conclusions, regardless of other new results or novel conclusions, I think the work is not internally self-consistent. It is my strong recommendation that the authors need to more carefully address this issue to be discussed below.*

We are glad to hear that we successfully addressed all the useful comments offered by the reviewer in their previous report, except one, which is concerned with the strong rise in resistivity with decreasing temperature. We are sorry for misunderstanding this point in our last response and we are grateful to the reviewer for their patience in clarifying the matter. We now understand their comment properly and can easily explain why we deal with the temperature dependence of the resistivity in the way we do, and why we consider it distinct from the effect of monopole scattering on the isothermal magnetoresistance. Our explanation is described in detail below.

The main issue here is, again, the theoretical interpretation of the resistivity. As summarized in the last sentence of Section II on page 11:

“Both effects lead to an electronic scattering rate, and hence a change in resistivity, proportional to the monopole density and sufficiently strong to account for the experimentally observed magnetoresistance”.

Coupled with their discussion in Supplemental Section S7, the authors claim that the observed change in resistivity is mainly related to “monopole density” (treating monopoles as effective scatterers). However, as I pointed out in previous report, this picture cannot explain the observation that resistivity also increases when the monopole density is reduced by lowering temperature. And the authors did not directly address this comment in their reply, they said:

“Our results (which deal with field-induced changes in monopole density at a fixed temperature) are not necessarily comparable to the temperature-dependence computed in the references.”

*Here the “references” are the two papers I mentioned in previous report. However, the point is that their interpretation of “field-induced changes in monopole density at a fixed temperature” is ***not*** consistent with their own experimental result shown in Fig. 1b and Fig. 2, which clearly shows the resistance increases at low temperatures (<~50K, which is also the regime of their field-dependence study). In particular, in their Fig. 2, for magnetic field larger than $\mu_0 H > \sim 0.5$ T, the monopole density $n_m(10K) > n_m(5K) > n_m(1.8 K)$ (shown in panel d). On the other hand, the resistance $R(10 K) < R(5K) < R(1.8 K)$ (shown in panel c), which shows the opposite trend.*

Again, we thank the reviewer for clarifying this point. We fully acknowledge that there are other contributions to the resistivity that are independent of the Ho magnetism and that they can have an even larger effect, as the reviewer notes in their comments. Indeed, we believe that **the main temperature-**

dependent contribution to the resistivity is unrelated to scattering and ought to be accounted for accordingly and not using a temperature-dependent scattering factor.

Resistivity in materials does not only depend on scattering rates, but also on the density of thermally available carriers. As we discuss in the introduction, $\text{Ho}_2\text{Ir}_2\text{O}_7$ undergoes a metal-insulator transition concomitant with Ir AIAO ordering, at ~ 80 K. This transition is naturally understood as caused by a change in electronic structure near the Fermi energy, **which is independent of Ho magnetism**. (That this is true is supported by the fact that all energy scales associated with Ho magnetism are far lower than 80 K, showing that Ho moments play no major role at this transition.) Like any insulator or semiconductor, the resistivity of $\text{Ho}_2\text{Ir}_2\text{O}_7$ increases dramatically with lowering temperature due to the **decreasing density of thermally available carriers, not because of increasing scattering**. The reduction in magnetic scattering at low temperatures counters this trend, but not strongly enough to change it qualitatively. At constant temperature, however, it is not expected that carrier density depends significantly on the magnetic field. On the other hand, the scattering rates contributed by the Ho magnetism are expected to change substantially under **isothermal changes in magnetic field**. This results in the observed hysteretic **magnetoresistance measured at constant temperature**, which our calculations indicate is controlled by the changing monopole density. We have now specified these points more clearly at several places in the manuscript to avoid any confusion for the reader.

The **abstract** has been altered to read: “we show that the *isothermal* magnetoresistance is highly sensitive to the monopole density.”

The following sentence on **Page 2** has been amended: “Measurements of the magnetic susceptibility and resistance show that for single crystals of $\text{Ho}_2\text{Ir}_2\text{O}_7$ the Ir ordering (Figure 1a) and MIT (Figure 1b) occur at approximately 80 K, *which is significantly higher than any energy scale associated with the magnetism of the Ho sublattice*.”

Caption to Figure 2 now reads: “*The temperature dependence of the resistivity is dominated by the insulating behaviour below the MIT, while the negative magnetoresistance at fixed temperature results from a combination of the paramagnetic response of spin ice at fixed monopole density and variations in the monopole density via the mechanisms described in the text.*”

Page 6 contains the amended sentence: “The temperature dependence of the zero-field resistance arises due to the insulating nature of $\text{Ho}_2\text{Ir}_2\text{O}_7$ below the MIT at 80 K *and is independent of the Ho magnetism.*”

Four sentences on **Pages 11 and 12** have been altered:

“We suggest that *the isothermal magnetoresistance* and monopole density are linked via two different scattering mechanisms that take place between the conduction electrons and the Ho magnetism at low temperatures.”

“Our results show that the *isothermal* magnetoresistance of $\text{Ho}_2\text{Ir}_2\text{O}_7$ is strongly linked to the concentration of magnetic monopoles...”

“Our results demonstrate that the hysteresis in magnetisation and *magnetoresistance* under an applied [111] magnetic field...”

“We note that the scattering mechanisms linking the monopole density and *the isothermal magnetoresistance* do not depend on the Ir magnetism.”

(By the way, it is a bit misleading that in their Fig. 3d, the authors shifted the “monopole-density vs H” curves for the T = 5K and 10K data downward by 70% and 90%.)

As explained above, there is a non-scattering contribution to the resistivity that dominates the temperature dependence in the regime of interest and that is essentially field-independent. There is no counterpart to account for it in the Monte Carlo simulations of the monopole density. Therefore, the monopole density curves ought to be compared to the resistivity curves with respect to their shape and up to some vertical offset (which is irrelevant to the results and conclusions presented in our work).

The **caption to Fig. 3(d)** has been amended accordingly: “Single-domain (100:0/0:100) curves are also shown. The 5 K (10 K) curves in (d) are shifted by 70% (90%) to allow for a clearer comparison of the

form of the monopole density and experimental magnetoresistance curves (the vertical offset is not relevant to the scattering mechanisms discussed in the text).”

Since the authors claim the resistivity is mainly affected by the “monopole density” (at least in the regime of their study), it shouldn’t matter whether the change of monopole density is due to H-field or temperature. The interpretation should work for both H- and T-effect.

The reviewer is correct in their claim. However, the temperature-dependent contribution to the resistivity, which (as discussed) is due to changing carrier density in an insulating material, significantly overpowers any contributions arising from the Ho moments. As a result, we cannot straightforwardly compare the temperature dependence of the monopole density and of the resistivity unless we quantitatively and accurately subtract the non-scattering term first. Fortunately, the insulator-like term is largely field-independent and therefore **the Ho moment contribution to the scattering dominates any changes in isothermal magnetoresistance**. We focus on isothermal field sweeps for exactly this reason and the experimental results and simulations support our conclusions about the effect of the monopole density in this regime.

It seems to me that in order for their theory to work (resistance is proportional to monopole density), one needs to introduce a strong temperature-dependent factor in the scattering theory. But no such dependence can be seen in Eq. (S4) or (S5) in the supplemental material. Another possibility is that other important scattering mechanism is not properly considered here. And if this is the case, it is unclear how much of their results can be solely attributed to monopole density.

We believe that the explanation in our response and the corresponding changes in the manuscript now completely address this important point without the need for introducing any additional exotic temperature-dependent quasiparticle scattering mechanisms. We thank the reviewer for stressing this potential source of confusion in our manuscript and for giving us the opportunity to improve our paper and further clarify our message.

Comments of Reviewer #2

I am grateful to the authors for fully addressing the comments in my previous review, and I believe the paper is suitable for publication in Nature Communications after one small change.

My only further suggestion is that the authors should include the details of their sample characterization (regarding stoichiometry) in either the methods or the SI. It will be useful for other workers in the field to understand what has been done in this regard -- the issue should be of broad interest. The description that they included in the referee response would be sufficient and could be simply pasted into the SI or Methods section with minimal rewording.

We thank the reviewer for their earlier useful comments and for recommending our manuscript for publication, barring one last simple addition.

In response to their remaining comment, we have added the following text in a **new section in the SI**:

“Potential issues with sample stoichiometry were mitigated at the synthesis stage as follows. Powder sample preparation: we prepared the phase-pure polycrystalline Ho₂Ir₂O₇ powder sample with an excess of 5% IrO₂ powder in order to compensate the evaporation loss due to high-temperature synthesis and used the final single-phase powder as a starting material for the crystal growth. Single crystal preparation: during the crystal growth process, which was carried out at relatively low temperature, we have tightly sealed a second outer crucible with alumina wool to catch any IrO₂ evaporation, but we did not observe any sign of it. The cell parameter values of both powder (10.1792 Å) and single crystals (10.1801 Å) are consistent and so we do not expect any variation in the chemical composition. To confirm this, we have annealed the as-grown crystal in oxygen atmosphere and found no change in the cell parameter value. Moreover, both Ho and Ir have stable valence states, 3+ and 4+ respectively, so we believe that any significant oxygen variation in the crystal is unlikely.”

“A qualitative estimate of the sample quality can be deduced from various physical properties. The measured lattice parameters of our crystals of Ho₂Ir₂O₇ are in excellent agreement with the structural trend across the RE₂Ir₂O₇ family, see Refs. [16, 17]. Furthermore, it has been previously argued [18] that the effect of oxygen vacancies in non-Kramers spin-ice materials is to suppress the moments on the rare-earth sites. If this was happening in our crystals of Ho₂Ir₂O₇, it would be detectable via experimental measurements of the saturation magnetization along different crystallographic directions. On the contrary, the results we obtain are in very good agreement with the values expected from calculations made with the full Ho³⁺ moment.”

We hope that this addresses all the reviewers' concerns and believe that the manuscript is now suitable for publication in Nature Communications.

REVIEWERS' COMMENTS

Reviewer #1 (Remarks to the Author):

I thank the authors for clarifying the issue of temperature dependence in this resubmission. I think this clarification is important, since otherwise the readers might get the impression that the electrical conductance is solely dominated by the electron-monopole scattering.

However, the “metal-insulator” transition (MIT) at approximately 80K in this compound is most likely of magnetic origin. For example, the long-range order of Ir-spins could lead to a gap-opening through the Slater mechanism. The MIT does not seem to be caused by the strong electron correlation. The authors claim that other mechanisms that lead to this MIT is mainly responsible for the temperature dependence of the resistivity.

However, given the fact that the MIT is most likely of magnetic origin, these “other mechanisms” will also be sensitive to magnetic field. As a result, these unidentified “other” scattering mechanisms could also contribute to the isothermal magnetoresistance. They might even be the dominant factor. For example, the resistivity could result from scattering of electrons off magnons (due to the Ir-ordering). And the magnon properties can be significantly altered by the magnetic field (given such a large change of magnetization considered in this work). Since the Ho-spins remain disordered, mechanisms related to the Coulomb phase, similar to those discussed in Udagawa et al. [PRL 108, 066406] and Chern et al. [PRL 110, 146602] might also be a factor, which also depend on the magnetic field.

While this work points out an interesting and potentially important mechanism (monopole-electron scattering) for the isothermal magnetoresistance in this new spin-ice compound, I don't think the study provides irrefutable evidences that the isothermal magnetoresistance is entirely due to monopole-scattering. In fact, other mechanisms might be even more important. As a result, the suggestion that the magneto-resistance can be used as a quantitative measure of monopole density is unclear. I suggest the authors explicitly discuss the limitations of their interpretation in the abstract and introduction and also discuss other possible mechanisms for isothermal magnetoresistance.

We thank the reviewer for engaging with our response to their earlier comments and with the resulting changes we made to the manuscript. We are pleased to see that the reviewer now recommends publication.

Reviewer's comment:

I thank the authors for clarifying the issue of temperature dependence in this resubmission. I think this clarification is important, since otherwise the readers might get the impression that the electrical conductance is solely dominated by the electron-monopole scattering.

However, the “metal-insulator” transition (MIT) at approximately 80K in this compound is most likely of magnetic origin. For example, the long-range order of Ir-spins could lead to a gap-opening through the Slater mechanism. The MIT does not seem to be caused by the strong electron correlation. The authors claim that other mechanisms that lead to this MIT is mainly responsible for the temperature dependence of the resistivity.

However, given the fact that the MIT is most likely of magnetic origin, these “other mechanisms” will also be sensitive to magnetic field. As a result, these unidentified “other” scattering mechanisms could also contribute to the isothermal magnetoresistance. They might even be the dominant factor. For example, the resistivity could result from scattering of electrons off magnons (due to the Ir-ordering). And the magnon properties can be significantly altered by the magnetic field (given such a large change of magnetization considered in this work). Since the Ho-spins remain disordered, mechanisms related to the Coulomb phase, similar to those discussed in Udagawa et al. [PRL 108, 066406] and Chern et al. [PRL 110, 146602] might also be a factor, which also depend on the magnetic field.

While this work points out an interesting and potentially important mechanism (monopole-electron scattering) for the isothermal magnetoresistance in this new spin-ice compound, I don't think the study provides irrefutable evidences that the isothermal magnetoresistance is entirely due to monopole-scattering. In fact, other mechanisms might be even more important. As a result, the suggestion that the magneto-resistance can be used as a quantitative measure of monopole density is unclear. I suggest the authors explicitly discuss the limitations of their interpretation in the abstract and introduction and also discuss other possible mechanisms for isothermal magnetoresistance.

Editor's suggestion:

In particular, following the comments of the reviewer, we believe it is important to include a paragraph in your discussion or concluding remarks, where the limitations of your interpretation are openly discussed.

Response:

We are happy to follow the reviewer's final suggestion of highlighting other transport mechanisms that may be at play coupling the applying field to the resistivity in this material. It gives us also the opportunity to further clarify a point where we politely disagree with the reviewer, namely that “other mechanisms may be even more important”. The Ho magnetic moments are nearly two orders of magnitude larger than the Ir moments. Any direct field coupling to the latter will be nearly two orders of magnitude weaker than the former, making it implausible for it to lead to a “more important” effect, as the referee claims. This was already explained in the manuscript, and the new paragraph gives us the opportunity to further clarify the point, whilst indeed adding a caveat about the limitations of our work, as suggested by the editor. We have added the following text to the final paragraph before the Conclusions (now relabeled as Discussion).

“Several mechanisms have previously been proposed to account for various transport properties of spin-ice iridates, especially Nd₂Ir₂O₇ and Pr₂Ir₂O₇ (see, e.g., [24–26]). While the mechanisms

proposed in these works may be at play in Ho₂Ir₂O₇ as well, the much stronger holmium moments mark out dipolar magnetic interactions, and the Coulomb field of spin-ice monopoles in particular, as a preeminent source of scattering in our case. This is also consistent with the close correlation between experimental magnetotransport and calculated monopole densities.”

[24] Udagawa et al. PRL 108, 066406 (2012).

[25] Chern et al. PRL 110, 146602 (2013).

[26] Ueda et al. PRL 115, 056402 (2015).

We believe that the paper is now ready for publication.

Kindest regards,

Claudio Castelnovo (on behalf of all the authors)